# Recent Advances in Microbial Production of *cis,cis*-Muconic Acid

**DOI:** 10.3390/biom10091238

**Published:** 2020-08-25

**Authors:** Sisun Choi, Han-Na Lee, Eunhwi Park, Sang-Jong Lee, Eung-Soo Kim

**Affiliations:** 1Department of Biological Engineering, Inha University, Incheon 22212, Korea; sisun95@gmail.com (S.C.); kamjjigina@naver.com (H.-N.L.); dmsgnl7241@naver.com (E.P.); 2STR Biotech Co., Ltd., Chuncheon-si, Gangwon-do 24232, Korea; lsj@strbiotech.co.kr

**Keywords:** *cis,cis*-Muconic acid, microbial production, shikimate pathway, metabolic engineering

## Abstract

*cis,cis*-Muconic acid (MA) is a valuable C6 dicarboxylic acid platform chemical that is used as a starting material for the production of various valuable polymers and drugs, including adipic acid and terephthalic acid. As an alternative to traditional chemical processes, bio-based MA production has progressed to the establishment of de novo MA pathways in several microorganisms, such as *Escherichia coli*, *Corynebacterium glutamicum*, *Pseudomonas putida*, and *Saccharomyces cerevisiae*. Redesign of the metabolic pathway, intermediate flux control, and culture process optimization were all pursued to maximize the microbial MA production yield. Recently, MA production from biomass, such as the aromatic polymer lignin, has also attracted attention from researchers focusing on microbes that are tolerant to aromatic compounds. This paper summarizes recent microbial MA production strategies that involve engineering the metabolic pathway genes as well as the heterologous expression of some foreign genes involved in MA biosynthesis. Microbial MA production will continue to play a vital role in the field of bio-refineries and a feasible way to complement various petrochemical-based chemical processes.

## 1. Introduction

*cis, cis-*Muconic acid (MA) is a six-carbon (C6) compound with two carboxylic functional groups at both ends and two double bonds in the middle [Figure 1]. A report published by Research and Market in 2019 showed that the turnover of the MA market was US$ 79.6 million in 2018, which is expected to be US$ 119.4 million in 2024 [1]. MA in the presence of catalysts can be converted to valuable industrial chemicals, such as adipic acid or caprolactam. In particular, adipic acid is a high-demand bulk intermediate chemical that can produce nylon-6,6, terephthalic acid (TPA), and polyurethanes. MA-driven intermediates, including adipic acid, are widely used in food additives, medicines, and cosmetics as well as in the textile industry [2]. On the other hand, the production of most of these polymers is petroleum-based. Specifically, the production of adipic acid produces numerous carcinogenic substances, such as cyclohexane, cyclohexanol, and cyclohexanone, via benzene, and causes environmental contamination problems. Recently, the production of benzene-free bio adipic acid through microbes has attracted attention to avoid these issues and meet the increasing demand for adipic acid. A key objective of related research is to secure economic feasibility by the production of high MA-producing microbial strains [2].

Draths and Frost (1994) first reported bio-based MA production. Moreover, the stable production of MA could be achieved in an environment-friendly manner by the establishment of a de novo biosynthetic pathway from 3-dehydroshikimate (DHS) in the shikimate pathway using glucose as a carbon source in *Escherichia coli* [2,3]. Since then, MA biosynthesis has been conducted by many researchers via re-engineering the shikimate pathway [2,3,4,5]. The shikimate pathway can be applied using three routes: (i) the pathway that synthesizes MA by the utilization of DHS as a precursor, (ii) the pathway with chorismate as a starting intermediate, and (iii) the biosynthesis of MA from aromatic amino acids (tyrosine, phenylalanine, and tryptophan). In addition, high MA productivity was induced through additional pathway engineering, such as glycolysis or the pentose phosphate pathway to send the carbon flux inside the cell to the shikimate pathway [2,3,4,5].

In research on microbial MA production, *E. coli* has been used representatively for many decades, including most research results and industrial feasibility studies. On the other hand, with the accumulation of various genetic and culture technologies, MA production utilizing various microbial strains with several merits has become possible, and remarkable progress has been achieved. Therefore, along with studies focusing on *E. coli*, this review paper also summarizes the production of bio-based MA with *Corynebacterium glutamicum*, *Klebsiella pneumoniae*, and *Pseudomonas putida*.

## 2. MA Production from *Escherichia coli*

### 2.1. MA Biosynthesis Through Redesign of the Shikimate Pathway

Previous studies used MA production via microbial strains with a simple one- or two-step process by the addition of compounds containing phenol rings, such as benzoate or catechol [4,5,6]. On the other hand, *E. coli* allowed the production of a real bio-based target MA through total biosynthesis using various pathways with glucose as a starter. Because the entire biosynthetic pathways that produce MA are not available in *E. coli*, it was necessary to develop a new pathway through the insertion of heterogeneous genes. Niu et al. reported a benzene-free microbial MA synthesis process that did not use carcinogenic benzene or benzene-derived chemicals as the problematic feedstock in the production of adipic acid [3]. In addition, this process did not cause any environmental problems, such as the generation of nitrous oxide that was a byproduct during MA synthesis. They used the *E. coli* AB2834 strain, in which *aroE* (encodes shikimate dehydrogenase) was replaced with leaky *aroE* to use DHS, which accumulates in the shikimate pathway. The MA biosynthesis pathway was then established in *E. coli* AB2834 through the heterologous expression of three genes. First, the *K. pneumoniae-*derived *aroZ* gene (encodes DHS dehydratase) was integrated into the chromosome; the *aroY* gene (encodes protocatechuate decarboxylase) derived from *K. pneumoniae*, and the *catA* gene (encodes Catechol 1,2-dioxygenase) derived from *Acinetobacter calcoaceticus* were expressed as the plasmids to generate *E. coli* WNl/pWN2.248. This engineered *E. coli* strain produced 36.8 g/L of MA from glucose [3]. Additional fermentation process optimization (Pathway 1 in Figure 1) resulted in an increase in the production yield to 59.2 g/L of MA [7].

Several studies have examined MA production via chorismate, which is another intermediate in the shikimate pathway. A pathway for the production of 2,3-dihydroxybenzoate (2,3-DHB) from chorismate was established by the expression of the genes that code EntC (encodes isochorismate synthase), EntB (encodes isochorismatase A), and EntA (encodes 2,3-dihydro-2,3-dihydroxybenzoate dehydrogenase). For two heterogeneous genes required to proceed from 2.3-DHB to MA, 2,3-DHB decarboxylase (EntX) from *K. pneumonia* and CatA from *P. putida* KT2440 were overexpressed using a high copy number plasmid. This enabled the production of 605 mg/L of MA from *E. coli*, in which the *aroG* (encodes DAHP synthase) and *aroL* genes (encoding shikimate kinase) were overexpressed to strengthen the shikimate pathway (Pathway 3 in Figure 1) [8,9].

In addition to DHS and chorismate, MA can also be produced by a pathway through catechol from anthranilate, which is the first intermediate of the tryptophan biosynthetic branch. Gram-per-liter levels of MA could be produced through the additional expression of the enzymes anthranilate 1,2-dioxygenase (ADO), which converts anthranilate to catechol and catechol 1,2-dioxygenase (CDO), which is engaged in the conversion of catechol to MA. In addition, 389.96 mg/L of MA was produced in a flask by the establishment of an anthranilate overproducer MA-4 strain. At the same time, tryptophan biosynthesis was blocked, and the key enzyme of the shikimate pathway was overexpressed (Pathway 5 in Figure 1) [10].

In addition, the MA biosynthesis pathway was divided into three in the presence of the starting intermediate of DHS and chorismate in *E. coli*. Each of these synthetic pathways were introduced to identify the best pathway to produce MA effectively [11]. Pathway 4 produced MA via chorismate, isochorismate, salicylate, and catechol in sequence, in which salicylate was converted to catechol by salicylate 1-monoxygenase and catechol was converted to MA in the presence of catechol 1,2-dioxygenase (Pathway 4 in Figure 1). Pathway 4 was designed based on the research showing that *E. coli* LS-8 produced 1.5 g/L of MA through the module engineering of these enzymes [12]. Pathway 2 was designed to produce MA from protocatechuate (PCA) through chorismate and 4-hydroxybenzoate. This was based on reports that 170 mg/L of MA was produced in minimal media prepared through engineered strains via the expression of three non-native genes (*pobA*, *aroY*, and *catA*, which code 4-hydroxybenzoate hydroxylase, protocatechuate decarboxylase, and catechol 1,2-dioxygenase, respectively) [13]. Pathway 1 was designed so that MA could be synthesized by PCA and catechol with the starting intermediate of traditional DHA. When the above three pathways were tested using an *E. coli* ATCC31882 derivative, which is an L-phenylalanine-overproducing strain, the traditional MA-producing Pathway 1 produced the highest yield of MA. The method involved fusing the protein that was engaged in the effective conversion of the intermediate. In addition, the fusion proteins of AroC (chorismate synthase) and MenF (isochorismate synthase) (Pathway 4 in Figure 1), and AroD (3-dehydroquinate dehydratase) and AroZ (DHS dehydratase) (Pathway 1 in Figure 1) were overexpressed to increase the carbon flux from chorismate to isochorismate. The level of MA production in the batch with pH control (with CaCO_3_) was 4.45 g/L. This proved that the traditional pathway to produce MA from DHS was still effective [11].

As another example of producing MA through the establishment of various pathways, several biosynthesis pathways were established in *E. coli*, and MA was produced via a biosynthesis route that was executed in the order of the DHS-derived route, chorismite-derived route, and tyrosine-phenol route (Pathway 6 in Figure 1). *E. coli* NST74 *ΔpheA ΔpykA ΔpykF Δcrr* (3.1 g/L minimal media) produced the highest MA production yield through the “metabolic funnel” (combination of Pathway 1 and 2), which is a parallel route to the combination of DHS and chorismite [14]. Choi et al. reported the highest production yield of MA in *E. coli*, which produced 64.5 g/L of MA by engineering the *E. coli* AB2834 strain pathway. This strain could produce up to 117 g/L (0.39 g/g) of the intermediate DHS. After establishing the heterogeneous *aroZ*, *aroY*, and *catA* (Pathway 1 in Figure 1) in the strain as the operon forms, a search was conducted for an efficient promoter based on the RNA-seq to express the heterogeneous genes. Finally, MA production of 64.5 g/L in 7-L fed-batch fermentation could be achieved, without accumulating intermediates in the *E. coli* strain [15].

### 2.2. MA Production via Various Carbon Sources and Co-Cultivation

Recently, the utilization of agricultural and industrial biomass as a feedstock in a bio-refinery has attracted increasing interest [16]. Of such materials, renewable lignocellulose materials promise an economically sustainable supply that is inexpensive and abundant. Lignin is a natural polymer in nature and a key component in woody plants. Recently, many studies have reported the use of lignin monomers and hemicellulose components rather than glucose or glycerol as carbon sources for the production of MA.

Lignocellulose, when pretreated, is degraded into glucose and xylose. Several studies have reported the use of these intermediates for MA production in *E. coli*. The Dahms pathway, which is the xylose degradation pathway, was introduced into *E. coli*. Two metabolically parallel pathways were then designed so that xylose was input directly into the TCA cycle as a carbon source through a catabolic pathway for growth. At the same time, glucose was input only into the shikimate pathway to produce MA. With the simultaneous input of the carbon sources of glucose and xylose, 4.09 g/L of MA was produced in minimal media [17].

A study on novel MA production was published in 2015 and involved two engineered *E. coli-E. coli* cocultures with glycerol as the carbon source. Two types of strains were established: (i) one strain produced high levels of DHS effectively by the deletion of *ydiB* (encoding quinate/shikimate dehydrogenase) and *aroE* (shikimate dehydrogenase) in a tyrosine overproducer *E. coli*, and (ii) in another strain, heterologous genes, *aroZ*, *aroY*, and *catA*, were inserted so that DHS could be converted to MA. The system was established to move the DHS effectively between the two *E. coli* strains, and ShiA (Shikimate transporter) permease, which is engaged in DHS assimilation, was overexpressed so that the secreted DHS could be absorbed efficiently into the cells. The two strains were cocultured in a bioreactor and produced 2 g/L of MA [18]. Moreover, a coculture of the two *E. coli* strains that used glucose or xylose with similar strategies provided an MA production of 4.7 g/L in the bioreactor [19].

These results suggest that MA could be biosynthesized through the engineering of numerous pathways in a single strain. On the other hand, it was also proven that coculture of engineered strains in accordance with the objective of each strain was equally effective.

## 3. MA Production from *C. Glutamicum*

Kinoshita et al. isolated a Gram-positive bacterium from the soil that produced large amounts of L-glutamate and called it *Micrococcus glutamicus*. Later, that strain was re-named *C. glutamicum*, and it has been used for the mass production of various amino acids, organic acids, polymer precursors, and biofuels. The strain is generally recognized as safe (GRAS) because of its characteristic of being a non-endotoxin. *C. glutamicum* is a representative industrial strain that produces L-glutamate and L-lysine, which are used as food additives with a production of more than 1.5 × 10^6^ tons/year and 0.9 × 10^6^ tons/year, respectively [20,21,22,23,24,25].

Because of the excellent characteristics of the strain for industry, many researchers have conducted studies on the production of useful materials from *C. glutamicum* through pathway engineering, and research on the strain to produce good quantities of MA have progressed, centering on the research by Becher et al. and Lee et al. [26,27]. Both groups removed the *catB* gene (encoding muconate cycloisomerase), which degraded MA from *C. glutamicum*, to accumulate MA as the final product in *C. glutamicum* (Figure 2).

Becher et al. produced MA from lignin-derived aromatic compounds, instead of glucose. When the lignin was treated hydrothermally in supercritical water, it hydrolyzed to aromatic compounds. The aromatic compounds entered the TCA cycle in the form of acetyl-CoA and succinyl-CoA, via catechol and the MA branch in the β-ketoadipate pathway inside *C. glutamicum* [28,29]. *C. glutamicum* devoid of *catB* in the β-ketoadipate pathway was prepared to accumulate MA in this pathway. Benzoic acid, phenol, and catechol were then added, which resulted in the accumulation of MA. In addition, the *catA* gene that coded catechol-1,2-dioxygenase, the final catalyst in MA biosynthesis by *C. glutamicum*, was replaced with a strong promotor. A fed-batch culture was then conducted by feeding the catechol on an hourly basis. The culture yielded 85 g/L of MA from catechol. The results showed that 1.8 g/L of MA was produced in the presence of aromatics when the hydrothermally depolymerized softwood lignin was applied in the engineered *C. glutamicum* strain [28].

Lee et al. engineered the shikimate pathway in *C. glutamicum* to produce MA from glucose. The *catB*, PCA dioxygenase alpha/beta subunit genes (*pcaG/H*), and shikimate dehydrogenase gene (*aroE*) were removed from the β-ketoadipate pathway to accumulate DHS and PCA, which were the precursors for MA production. Subsequently, to connect the missing conversion from PCA to catechol, the encoding PCA decarboxylase (*aroY*) gene derived from *K. pneumoniae* and the *kpdBD* (encoding PCA decarboxylase subunit) gene, which was a subunit of AroY, were codon-optimized in *C. glutamicum* so MA would accumulate. This resulted in MA production (340 mg/L) from glucose. The established strain could produce 53.8 g/L of MA through 50-L scale fed-batch fermentation and media optimization [29]. The uptake of glucose by *C. glutamicum* through the PTS system results in the consumption of phosphoenolpyruvate (PEP), an important starter of the shikimate pathway. The aim was to reduce PEP consumption by removing phosphotransferase (*ptsl*) from the PTS system. On the other hand, although the glucose uptake rate was reduced five-fold compared to the parental strain, cell growth was retarded. This problem was resolved by strengthening the inositol permease transporter, which is another glucose uptake system, by removing the *iolR* gene, which encoded the IolR repressor that repressed the inositol permease transporter. In addition, the *qsuB* gene (encoding 3-dehydroshikimate dehydratase), which converted DHS (one of the essential pathways for MA production) to PCA, was overexpressed along with YBD (*aroY* and *kpdBD*), after which 4.5 g/L of MA was produced, showing a 12.2% increase [30,31].

From the above results, a possible MA production pathway from glucose using a *C. glutamicum* strain with a well-established amino acid production system was proposed as an industrial-scale pilot. In addition, the strain tolerates aromatic compounds, including the precursors (PCA, catechol) contained in *C. glutamicum*, proving that it could be a beneficial strain for high-level MA production.

## 4. MA Production from Other Microorganisms

Some microorganisms belonging to the genera *Pseudomonas*, *Arthrobacter*, *Corynebacterium*, *Brevibacterium*, *Microbacterium*, and *Sphingobacterium* were reported to metabolize benzoate via the catechol branch of the β-ketoadipate pathway to produce MA.

Benzoate is first converted to benzoate diol catalyzed by benzoate 1,2-dioxygenase encoded by *benABC.* The oxidative decarboxylation of benzoate diol to catechol is then performed by benzoate diol dehydrogenase encoded by *benD*. Ring fission of catechol between the hydroxyl groups is catalyzed by CatA encoded by *catA* to form MA. The latter metabolite is then converted to muconolactone by muconate cycloisomerase encoded by *catB*. Muconolactone is finally converted to tricarboxylic acid cycle intermediates after several metabolic steps.

### 4.1. MA Production from Saccharomyces Cerevisiae

*E. coli* strains that can only grow under neutral pH conditions are not beneficial in cost-competitive industrial production processes because, even if they are improved to high MA producing strains, the MA is purified under low pH conditions. In contrast, *Saccharomyces cerevisiae* is beneficial in industrial production because it can be fermented at a low pH, has high robustness, is resistant to toxic inhibitors and fermentation products, has microbial contamination resistance, and has a high level of public acceptance [32].

Weber C et al. reported the results for MA production by introducing the heterologous biosynthetic pathway from DHS using *S. cerevisiae*. As with the other strains, a three-step synthetic pathway (dehydroshikimate dehydratase from *Podospora anserina*, protocatechuic acid decarboxylase from *Enterobacter cloacae*, and catechol 1,2-dioxygenase from *Candida albicans*) was introduced into the yeast, and further genetic modification and feedback inhibition mitigation were applied. The *S. cerevisiae* MuA12 strain, where the precursor availability was enhanced, could produce 141 mg/L of MA in a flask culture [33].

Similarly, three enzymes, AroZ from *Podospora anserina*, AroY from *K. pneumoniae*, and HQD2 from *Candida albicans*, were introduced to a multicopy plasmid, and the transketolase gene (TKL1) was overexpressed for carbon flux. The resulting system produced 559.5 mg/L of MA from glucose through the engineered *S. cerevisiae* MuA12 [34].

In addition to rational engineering for MA production, a combined adaptive laboratory evolution (ALE) strategy and rational metabolic engineering were also employed. An *S. cerevisiae* strain that produced more aromatic amino acid was secured using exogenous amino acid supplementation (particularly, tryptophan) and anti-metabolite selection [4-fluorophenylalanine and G418 (antibiotic)] methods, and 2.1 g/L of MA was produced from the MuA-5.01.1.02+aro1t+scPAD1 strain in fed-batch fermentation [35].

In addition to an engineering strategy at the metabolic site, such as gene deletion or overexpression of the structural genes, another study produced MA and shikimate from *S. cerevisiae* by removing Ric1, which is a transcriptional repressor. In silico modeling and pathway analysis confirmed the production of 2705 mg/L of MA and its precursor using the BY47471-MA4 strain [36].

Recently, it was reported that *S. cerevisiae* could produce MA from lignocellulosic biomass hydrolysate using recombinant xylose-fermenting yeast. In addition to the exogenous MA biosynthetic pathway, the xylose isomerase gene from *Bacteroides vulgatus* and pentose phosphate pathway genes from *S. cerevisiae* were overexpressed in the yeast strain, and the overexpression of the Aro1 gene (with a stop codon of AroE) and a feedback-resistant Aro4^opt^ mutant gene from *S. cerevisiae* were also applied. Under aerobic conditions, the MA titer reached 424 mg/L, and 1286 mg/L MA was produced with the supplement of 1 g/L catechol. Fermentation of an oil palm empty fruit bunch hydrolysate resulted in 31.3 g/L ethanol and 53.4 mg/L MA [37].

### 4.2. MA Production from Amycolatopsis Species

*Amycolatopsis* spp. also tolerate lignin-based aromatics, such as catechol, guaiacol, phenol, toluene, p-coumarate, and benzoate, and even favor these aromatics as carbon sources over sugar. The aqueous phase, obtained through hydrothermal conversion as a lignin treatment, contained a large quantity of guaiacol of 7 g/L. Research on the metabolically engineered *Amycolatopsis* sp. ATCC 39,116 revealed 3.1 g/L of MA production from the guaiacol, while 1.8 mM of MA was produced with the lignin hydrolysate [38].

### 4.3. MA Production from Pseudomonas Species

*P. putida* has excellent tolerance towards organic solvents and was the first soil microbe among Gram-negative bacteria to be recognized as a safety strain from the Recombinant DNA Advisory Committee [39,40,41]. Therefore, *P. putida* has attracted attention as a metabolic engineering chassis for applications in the industrial bioengineering field. Moreover, *P. putida* can be used as a carbon source for the growth and energy production of lignin-related aromatics, such as vanillin [42,43,44,45].

Sonoki et al. conducted an experiment on *P. putida* to produce MA from lignin without glucose through metabolic engineering. The precursor was induced to accumulate by the removal of *pcaG/H* and *catB* from the β-ketoadipate pathway using the same strategy as with *C. glutamicum*. The PCA decarboxylase gene was overexpressed to complete the MA pathway. A medium containing 25 nM each of aromatic compound 4-hydroxybenzoic acid (4-HBA) and vanillin produced MA with a yield of 19.0%, whereas 0.11 mM MA was biosynthesized through softwood lignin [46,47].

Vardon et al. substituted protocatechuate 3,4 dioxygenase (PcaG/H) with protocatechuate decarboxylase (AroY) to convert aromatics from the catechol and protocatechuate branches to MA with *P. putida* KT2440. To block the synthesized MA, *P. putida* KT2440-CJ103 was established, in which a *catBCA* operon (metabolism operon of MA through β-ketoadipate pathway) was substituted with Ptac_*catA_dmpKLMNOP* (encoding phenol monooxygenase), and 13.5 g/L of MA was produced from the p-coumarate through fed-batch fermentation conducted for 78.5 h [48].

Two subunit genes, which play important roles in PCA decarboxylase and decarboxylase activity, were expressed as a genome-integrated gene to reduce PCA accumulation, which is a bottleneck in MA production, resolve the conversion of PCA to MA, and increase MA production. The KT2440-CJ184 (*P. putida* KT2440 Δ*catRB*C::Ptac:*catA* Δ*pcaHG*::Ptac:*aroY*:*ecdB*:*ecdD*) strain expressed a codon-optimized AroY, EcdB (subunit of decarboxylase AroY) and EcdD (subunit of decarboxylase AroY) from *E. cloacae*, and produced 15.59 g/L of MA in bioreactor cultivation that contained p-coumarate, which is an aromatic lignin monomer, and glucose [49]. This was followed by the accumulation of 4-hydroxybenzoate (4-HBA) and vanillin, which are intermediates of PCA in the same group. The catabolite repression control (Crc) protein, which is a regulator of carbon catabolite repression, was located and deleted. This increased the MA conversion rate from p-coumarate by 12% [50].

Furthermore, a pathway was engineered that could catabolize a range of aromatic compounds based on *P. putida* KT2440 and could convert them to 16 catabolic intermediates, which exhibited a substantive chemical diversity. Enzymes derived from *Sphingobacterium* sp., *Paenibacillus* sp., and *P. putida* were introduced to redesign the aromatic catabolic pathway so that these catabolic-intermediate molecules could be produced from an aromatic compound or glucose. The 16 target molecules were then produced and analyzed based on a bioreactor. Among the numerous mutants produced in this study, the *P. putida* KT2440 Δ*catRBC*::Ptac:*catA* Δ*pcaHG*::Ptac:*aroY:ecdB:asbF* Δ*pykA*::*aroGD*146N:*aroY*:*ecdB*:*asbF ΔpykF Δppc Δpgi-1 Δpgi-2 Δgcd* (CJ522) strain was used to produce MA; 12 g/L of MA was produced through a fed-batch mode bioreactor cultivation from glucose. This is a practical example of the use of microbial strains for the production of chemically diverse molecules as building blocks [51].

Although the β-ketoadipate pathway is not present in *E. coli* or *S. cerevisiae*, it is endogenous to aromatic degrading organisms, such as *P. putida*. This is because *P. putida* has been engineered extensively to produce MA directly from pretreated biomass as well as from lignin-derived monomers. Table 1 shows the list of engineered microbial strains for MA biosynthesis.

## 5. Conclusions and Perspectives

Recently, information on the metabolic pathways of numerous microbes has become readily available, owing to the development of extensive genomic analysis, well-established molecular tools, system biology techniques, and fermentation techniques. Therefore, with such technologies, it has become possible to build new metabolisms that did not exist in nature or to reinforce existing ones. MA synthesis via PCA and catechol as starters in the shikimate pathway has been applied to various microbes, starting from primary research with *E. coli*.

In addition, MA production with *E. coli* has been diversified with microbes, such as *C. glutamicum* and *P. putida*, which have resistance to the toxicity of aromatic compounds. These aromatic-tolerant microbes possess abundant pathways and enzymes that can produce MA with aromatic compounds that are being discarded. This paper summarizes the use of microbes with strong potential as microbial cell factories at an industrial level for MA production. Finally, various other aromatics should be applied to obtain higher MA titers considering the economic and industrial aspects in the mass production of MA and the metabolic engineering approaches and optimized process operations using a range of microbes.

## Figures and Tables

**Figure 1 biomolecules-10-01238-f001:**
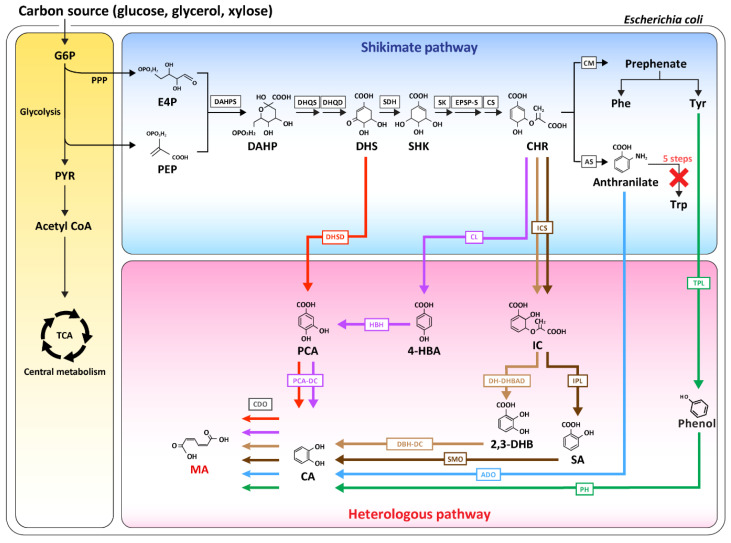
Schematic overview of the metabolic pathway for MA biosynthesis in *E. coli*. The red arrow (pathway 1), purple arrow (pathway 2), tan arrow (pathway 3), brown arrow (pathway 4), blue arrow (pathway 5), and green arrow (pathway 6) represent MA biosynthetic pathways 1, 2, 3, 4, 5, and 6, respectively. G6P, glucose-6- phosphate; PPP, pentose phosphate pathway; PYR, pyruvate; PEP, phosphoenolpyruvate; E4P, erythrose4-phosphate; DAHP, 3-deoxy-D-arabinoheptulosonate-7-phosphate; DHS, 3-dehydroshikimate; SHK, shikimic acid; Phe, phenylalanine; Tyr, tyrosine; Trp, tryptophan; CHR, chorismate; 4-HBA, 4-hydroxybenzoate; IC, isochorismate; 2,3-DHB, 2,3-dihydroxybenzoate; SA, salicylate; PCA, protocatechuate; CA, catechol; MA, muconic acid. DAHPS, DAHP synthase; DHQS, DHQ synthase; DHQD, DHQ dehydratase; SDH, shikimate dehydrogenase; SK, shikimate kinase; EPSP-S, 5-enolpyruvylshikimate-3-phosphate synthase; CS, chorismate synthase; CM, chorismate mutase; AS, anthranilate synthase; DHSD, DHS dehydratase; PCA-DC, protocatechuate decarboxylase; CDO, catechol 1,2-dioxygenase; TPL, tyrosine phenol lyase; PH, phenol hydroxylase; CL, chorismate pyruvate-lyase; ICS, isochorismate synthase; IM, isochorismatase; DH-DHBAD, 2,3-dihydro-2,3-DHBA dehydrogenase; IPL, isochorismate pyruvate lyase; DBH-DC, 2,3-dihydroxybenzoate decarboxylase; SMO, salicylate 1-monoxygenase; ADO, anthranilate 1,2-dioxygenase.

**Figure 2 biomolecules-10-01238-f002:**
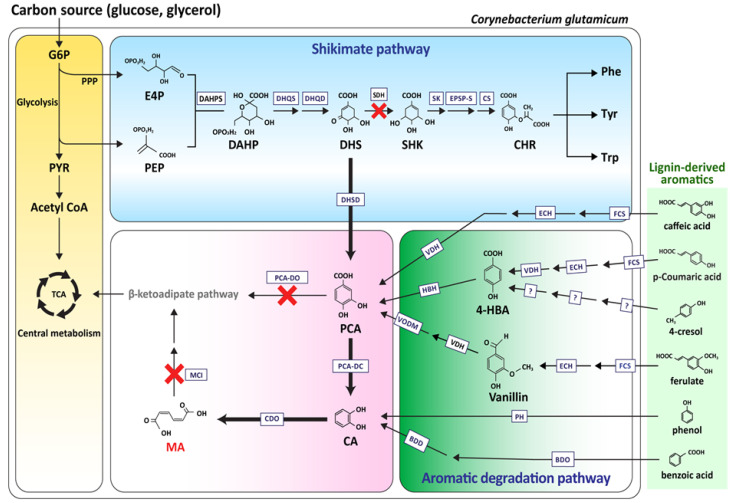
Schematic overview of the aromatic compound degradation metabolic pathway in *C. glutamicum* for MA biosynthesis. G6P, glucose-6- phosphate; PPP, pentose phosphate pathway; PYR, pyruvate; PEP, phosphoenolpyruvate; E4P, erythrose4-phosphate; DAHP, 3-deoxy-D-arabinoheptulosonate-7-phosphate; DHS, 3-dehydroshikimate; SHK, shikimic acid; Phe, phenylalanine; Tyr, tyrosine; Trp, tryptophan; CHR, chorismate; 4-HBA, 4-hydroxybenzoate; IC, isochorismate; 2,3-DHB, 2,3 dihydroxybenzoate; SA, salicylate; PCA, protocatechuate; CA, catechol; MA, muconic acid. DAHPS, DAHP synthase; DHQS, DHQ synthase; DHQD, DHQ dehydratase; SDH, shikimate dehydrogenase; SK, shikimate kinase; EPSP-S, 5-enolpyruvylshikimate-3-phosphate synthase; CS, chorismate synthase; DHSD, DHS dehydratase; PCA-DC, protocatechuate decarboxylase; CDO, catechol 1,2-dioxygenase; MCI, muconate cycloisomerase; PCA-DO, protocatechuate 3,4-dioxygenase; FCS, feruloyl–CoA synthetase; ECH, enoyl–CoA hydratase/aldolase; VDH, vanillin dehydrogenase; HBH, p-hydroxybenzoate hydroxylase; VODM, vanillate O-demethylase: PH, phenol hydroxylase; BDO, benzoate dioxygenase; BDD, benzoate diol dehydrogenase.

**Table 1 biomolecules-10-01238-t001:** List of engineered microbial strains for MA biosynthesis.

Organism/Strain	Feedstock	Titer (g/L)	Culture Method (Working Volume)	Fermentation Duration (h)	References
*Escherichia coli*	Glucose	36.8	Fed-batch (2 L)	48	[3]
*E. coli*	Glucose	0.61	Batch (50 mL)	72	[9]
*E. coli*	Glucose, glycerol	0.39	Batch	32	[10]
*E. coli*	Glucose	4.45	Batch	72	[11]
*E. coli*	Glucose, glycerol	1.5	Batch	48	[12]
*E. coli*	Glucose	0.17	Batch (50 mL)	72	[13]
*E. coli*	Glucose, glycerol	64.5	Fed-batch (2 L)	120	[15]
*E. coli*	Glucose	4.09	Batch (10 mL)	72	[17]
*E. coli*	Glycerol	2	Batch (1 L)	72	[18]
*E. coli*	Glucose, xylose	4.7	Fed-batch	72	[19]
*Corynebacterium glutamicum*	Glucose, catechol	85	Fed-batch (300 mL)	60	[26]
*C. glutamicum*	Glucose	38	Fed-batch (3 L)	-	[27]
		54	Fed-batch (18 L)		
*C. glutamicum*	Glucose	4.5	Batch (5 mL)	72	[31]
*Saccharomyces cerevisiae*	Glucose	0.00156	Batch (50 mL)	170	[32]
*S. cerevisiae*	Glucose	0.14	Batch	108	[33]
*S. cerevisiae*	Glucose	0.56	Batch	96	[34]
*S. cerevisiae*	Glucose	0.56	Batch (30 mL)	-	[35]
		2.1	Fed-batch (1.7 L)	240	
*S. cerevisiae*	Glucose	0.32	Batch (25 mL)	72	[36]
*S. cerevisiae*	Glucose, xylose, catechol	1.29	Batch (50 mL)	79	[37]
*S. cerevisiae*	Glucose, catechol	~1	96-well shake plates	72	[52]
*S. cerevisiae*	-	0.43	24-deep well plates (3 mL)	72	[53]
*S. cerevisiae*	Glucose	0.86	Batch	192	[54]
		1.2	Fed-batch	168	
	Glucose + amino acids	5.1			
*S. cerevisiae*	Glucose	1.24	Batch (50 mL)	144	[55]
*S. cerevisiae*	Glucose	0.8	24-deep well plates (3 mL)	72	[56]
*S. cerevisiae*	Glucose	20.8	Fed-batch (1.3 L)	149.5	[57]
*Amycolatopsis* sp. ATCC39116	Guaiacol	3.1	Fed-batch (100 mL)	24	[38]
*Pseudomonas putida*	Benzoate	34.5	Fed-batch (8 L)	124	[44]
*P. putida*	Vanillic acid, 4-hydroxybenzoic acid	1.35	Batch (5mL)	-	[47]
*P. putida*	p-coumarate	13.5	Fed-batch (700 mL)	78.5	[48]
*P. putida*	p-coumarate	15.6	Fed-batch(300 mL)	73	[49]
*P. putida*	Glucose	4.92	Fed-batch (300 mL)	54	[49]
*P. putida*	Glucose	12	Fed-batch	144	[51]
*P. putida*	Glucose, benzoate	32	Fed-batch	40	[58]
*P. putida*	Glucose, benzoate	18.5	Fed-batch (3.5 L)	56	[59]
*Pseudomonas* sp. 1167	Benzoate, succinate	7.18	Batch (30 mL)	11	[60]
*P. putida*	Benzoate	52.3	Fed-batch (300 mL)	-	[61]
*P. putida*	Catechol	64.2	Fed-batch (500 mL)	62	[62]
*Pseudomonas* sp. NGC7	Vanillate	3.2	Fed-batch (200 mL)	72	[63]
*P. putida*	Glucose	22	Fed-batch	104	[64]
*Arthrobacter* sp. T8626	Benzoate	27	Fed- batch (300 L)	69	[65]
*Sphingobium* sp. SYK-6	Vanillic acid, syringaldehyde, syringic acid	0.027	Fed-batch (10 mL)	48	[47]
*E. coli*	Catechol	59	Fed-batch	12	[66]
*E. coli*	Glucose	1.53	Batch	36	[67]
*Klebsiella pneumoniae*	Glucose	2.1	Batch (50 mL)	72	[68]

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
