# Peer review of "Recent Advances in Microbial Production of cis,cis-Muconic Acid"

_biomolecules, 2020, doi:10.3390/biom10091238_

Round 1

Author Response

Reviewer 1.

On the manuscript entitled "Recent Advances in Microbial Production of cis,cis-Muconic Acid” for Biomolecules The manuscript represents a comprehensive, informative and easy-to-follow to read review on highly attractive subject of microbial cis,cis-muconic acid production by. cis,cis-Muconic acid covered by the manuscript is the compound with high demands on the market, which assures interest of broad audience. The Authors logically organized each of the paragraphs - describing the market of the compound of interest, then metabolic pathways of biosynthesis of cis,cismuconic acid by Escherichia coli, Corynebacterium glutamicum, Pseudomonas putida, and Saccharomyces cerevisiae. Cited literature cover the most relevant and also very recent papers, important in the field. Illustrations are clear and informative - assist comprehension of the text. It is hard to indicate any serious flaws in the manuscript. I indicated only minor remarks.

- Thank you very much for your positive comment

Page 2, line 56-59 - Please specify the cited literature.

- As suggest, the references [2, 3, 4, 5] were included.

Page 3, line 70-72 - Please correct the font size.

- As suggested, the font size was corrected.

Page 3, line 80 - Specify reference number

- As suggested, the reference number [80] was included.

Reviewer 2 Report

This manuscript of the review (Si-Sun Choi et al.) describes about the microbial production of cis,cis-muconic acid by Escherichia coli, Pseudomonas putida, Corynebacterium glutamicum, and others. The manuscript gives valuable information, but there are some points to be improved as follows.

Some parts of Table 1 and the main text do not match, and the entire manuscript needs to be checked carefully.

Figure 2: This is the aromatic degradation pathway for C. glutamicum.

How about in the case of P. putida?  Any differences of the aromatic degradation pathway between C. glutamicum and P. putida?

Table 1 referenced majorly about P. putida than C. glutamicum related to the aromatic degradation pathway.

p.4, line 136-142: Choi et al. reported high production of MA (64.5 g/L).

It should be shown in Table 1.

The description in the text does not match Table1 as follows.

(mistake for reference number? Please check the other parts also.)

p.5, line 165: 2 g/L of MA [18],

p.5, line 166-167: MA production of 4.7 g/Lin the bioreactor [19]

p.8, line 270:  2,362 mg /L of MA…[36].

    The abstract of reference No. 36 describes 2.7 g/L of MA.

    Please check the titer and feedstock.

The data of reference No. 17 should be appeared in Table1.

Table 1 S. sp. SK-6:  Sphingobium sp. SYK-6

(I couldn’t find abbreviations in the text.)

Some words appeared with different font and size:

 P.1, line 13 : Escherichia, Corynebacterium  (different size)

 p.2, line 50-51: Protocatechuate decarboxylase (different font)

Author Response

Reviewer 2.

This manuscript of the review (Si-Sun Choi et al.) describes about the microbial production of cis,cis-muconic acid by Escherichia coliPseudomonas putida, Corynebacterium glutamicum, and others. The manuscript gives valuable information, but there are some points to be improved as follows.

Some parts of Table 1 and the main text do not match, and the entire manuscript needs to be checked carefully.

- Thank you very much for your positive comments. According to your suggestion, we checked the entire manuscript along with the Table 1.

Figure 2: This is the aromatic degradation pathway for C. glutamicum.

How about in the case of P. putida?  Any differences of the aromatic degradation pathway between C. glutamicum and P. putida? Table 1 referenced majorly about P. putida than C. glutamicum related to the aromatic degradation pathway.

- Since both C. glutamicum and P. putida use the similar aromatic degradation pathway, we decided to focus on Corynebacteria which produce much higher titer of muconic acid. Since many MA low-titer Pseudomonas cases were reported from the literature review, we decided to include all the reported cases in the Table 1. Due to the limitation of the main text, we could not describe all the cases listed in the table 1, rather to focus on the high priority cases.

p.4, line 136-142: Choi et al. reported high production of MA (64.5 g/L). It should be shown in Table 1.

- As suggested, it was included in Table 1.

The description in the text does not match Table1 as follows. (mistake for reference number? Please check the other parts also.)

p.5, line 165: 2 g/L of MA [18], p.5, line 166-167: MA production of 4.7 g/Lin the bioreactor [19]

- As suggested, all the mismatches were corrected.

p.8, line 270:  2,362 mg /L of MA…[36]. The abstract of reference No. 36 describes 2.7 g/L of MA. Please check the titer and feedstock.

- The 2.7 g/L titer reported in the ref. 36 was the sum of MA and its precursor PCA. The MA-only titer was 0.32 g/L listed in the Table 1. The line 270 was revised as following; ‘In silico modeling and pathway analysis confirmed the production of 2,705 mg/L of MA and its precursor using the BY47471-MA4 strain [36]. 

The data of reference No. 17 should be appeared in Table1.

- As suggested, the ref. 17 data were included in Table 1.

Table 1 S. sp. SK-6:  Sphingobium sp. SYK-6 (I couldn’t find abbreviations in the text.)

- The S. sp. SK-6 was replaced with the full name, Sphingobium sp. SYK-6 in Table 1.

Some words appeared with different font and size:

P.1, line 13 : Escherichia, Corynebacterium  (different size)

p.2, line 50-51: Protocatechuate decarboxylase (different font)

- We checked and corrected all the fonts and sizes, which might happened during the publication processing.

Round 2

Reviewer 2 Report

This manuscript of the review (Si-Sun Choi et al.) describes about the microbial production of cis,cis-muconic acid by Escherichia coli, Pseudomonas putida, Corynebacterium glutamicum, and others.  The manuscript gives valuable information, but there are some points to be improved as follows.

Some parts of Table 1 and the main text STILL do not match, and the entire manuscript needs to be checked carefully again.

The description in the text does not match with Table1 as follows.

p.4, line 129-130: 4.45 g/L of MA [11],

but this is missed in Table1.

p.8, line 285 :  The reference number [38] is different from Table 1([65]).

p.9, line 309-310:  15.59 g /L of MA…[49].

but this value is different from Table 1 in page 11 (4.92 g/L MA from glucose [49]).

Author Response

This manuscript of the review (Si-Sun Choi et al.) describes about the microbial production of cis,cis-muconic acid by Escherichia coli, Pseudomonas putida, Corynebacterium glutamicum, and others.  The manuscript gives valuable information, but there are some points to be improved as follows.

Some parts of Table 1 and the main text STILL do not match, and the entire manuscript needs to be checked carefully again.

The description in the text does not match with Table1 as follows.

p.4, line 129-130: 4.45 g/L of MA [11], but this is missed in Table1.

- As suggested, ref. 11 was included in the table 1.

p.8, line 285 :  The reference number [38] is different from Table 1([65]).

- We had a mistake. The ref. 38 and ref 65. were the same one. So, we deleted the ref. 65 and re-numbered the ref.66-69 to ref. 65-68.

p.9, line 309-310:  15.59 g /L of MA…[49].

but this value is different from Table 1 in page 11 (4.92 g/L MA from glucose [49]).

- Since the ref. 49 showed two results (e.g. glucose feeding and p-coumarate feeding), the p-coumarate feeding result of 15.6 g/L was listed in the text and the glucose feeding result of 4.92 g/L was listed in the table 1. To avoid this confusion, we decided to list both results in the table 1.